# In Silico Drug Repurposing Approach: Investigation of *Mycobacterium tuberculosis* FadD32 Targeted by FDA-Approved Drugs

**DOI:** 10.3390/molecules27030668

**Published:** 2022-01-20

**Authors:** Nolwazi Thobeka Portia Ngidi, Kgothatso Eugene Machaba, Ndumiso Nhlakanipho Mhlongo

**Affiliations:** School of Laboratory Medicine and Medical Sciences, University of KwaZulu-Natal, Durban 4001, South Africa; 215022865@stu.ukzn.ac.za (N.T.P.N.); machabak@ukzn.ac.za (K.E.M.)

**Keywords:** *Mtb*-FadD32, drug repurposing, MD simulations, post-MD analysis

## Abstract

**Background**: Despite the enormous efforts made towards combating tuberculosis (TB), the disease remains a major global threat. Hence, new drugs with novel mechanisms against TB are urgently needed. Fatty acid degradation protein D32 (FadD32) has been identified as a promising drug target against TB, the protein is required for the biosynthesis of mycolic acids, hence, essential for the growth and multiplication of the mycobacterium. However, the FadD32 mechanism upon the binding of FDA-approved drugs is not well established. Herein, we applied virtual screening (VS), molecular docking, and molecular dynamic (MD) simulation to identify potential FDA-approved drugs against FadD32. **Methodology/Results**: VS technique was found promising to identify four FDA-approved drugs (accolate, sorafenib, mefloquine, and loperamide) with higher molecular docking scores, ranging from −8.0 to −10.0 kcal/mol. Post-MD analysis showed that the accolate hit displayed the highest total binding energy of −45.13 kcal/mol. Results also showed that the accolate hit formed more interactions with FadD32 active site residues and all active site residues displayed an increase in total binding contribution. RMSD, RMSF, Rg, and DCCM analysis further supported that the presence of accolate exhibited more structural stability, lower bimolecular flexibility, and more compactness into the FadD32 protein. **Conclusions**: Our study revealed accolate as the best potential drug against FadD32, hence a prospective anti-TB drug in TB therapy. In addition, we believe that the approach presented in the current study will serve as a cornerstone to identifying new potential inhibitors against a wide range of biological targets.

## 1. Introduction

Tuberculosis (TB) is an ancient infectious disease caused by the pathogenic bacillus *Mycobacterium tuberculosis* (*Mtb*); this disease is a major global health problem and is one of the top 10 causes of death worldwide [1]. The 2019 global statistics of TB reported, approximately 10 million people developed TB and approximately 1.4 million TB deaths occurred [1]. Currently, the approved first-line drugs for the treatment of TB are isoniazid (INH), rifampicin (RIF), pyrazinamide (PZA), and ethambutol (EMB) for a period of 6 to 9 months [2]. However, there is frequent emergence of strains that are resistant to these drugs, these strains include: multi-drug-resistant (MDR), extensively-drug-resistant (XDR), and total-drug-resistant (TDR) strains [3,4].

Drug-resistant TB is hindering progress towards combating TB and half a million people developed rifampicin-resistant TB and 78% had MDR TB in 2019 [1]. The sudden emergence of the COVID-19 pandemic has caused the situation to worsen, consequently, reversing the progress made towards reducing the burden of TB [1]. In recognition of the enormous health, economic and social impacts posed by this disease on the public, research breakthroughs are needed to rapidly reduce TB cases [5]. Therefore, there is a demand for further research into the discovery of new drug targets, and new potent inhibitors with novel mechanisms of action against TB [6].

Mycolic acid is a major component of the bacterial cell wall, the thick waxy coat trait renders protection to the bacteria, hence, providing immunity against the host’s immune system and current antibiotics [7,8]. The mycolic acid biosynthesis pathway is well-validated and used to identify potential targets for antimycobacterial drug development [9]. Fatty acid degradation protein D32 (FadD32) is required for the biosynthesis of mycolic acids, hence, essential for the growth and multiplication of the mycobacterium [10]. FadD32 is a bifunctional enzyme, it initially catalyzes meromycolic acid and ATP to form meromycoloyl-AMP, and then catalyzes the acyl chain transfer from meromycoloyl-AMP to the phosphopantetheinyl arm of the N-terminal ACP domain of Pks13 [10].

FadD32 (Figure 1) has been identified and proposed as a novel drug target for the development of potential drugs against TB [10,11]. The structure comprises two distinct domains, the N-terminal domain (1–483 residues) and the C-terminal domain (484–630 residues) [10]. FDA-approved drugs against FadD32 remain a mystery in literature. Hence, the FadD32 mechanism is not well established. Therefore, unknown FDA-approved drug against FadD32 has motivated the search for new potent FDA-approved drugs (drug repurposing) against FadD32, a novel potential target.

Repurposing FDA-approved drugs is an effective, time-saving, and less expensive strategy to discover new molecules against drug targets of interest [12]. Hence, it is uncomplicated to develop the drugs since essential data about them is already available, which minimizes the risk of failure [13]. Repurposed drugs used in TB treatment include moxifloxacin, linezolid, clofazimine, amikacin, and meropenem [14,15], these drugs have proven to be effective against MDR-TB and XDR-TB [15,16]. Inhibitors that have the potential to target multiple drug targets/sites within the mycobacterial cell are imperative. To achieve the above, different computational methods can be applied [17].

Virtual screening (VS) has proven to be a powerful and indispensable tool in discovering small molecular inhibitors which bind to drug targets of interest [18]. The computer-aided technique screens large libraries of compounds, filters, and discards undesirable compounds, hit compounds are identified and selected as lead compounds [19]. This work focuses on structure-based VS, a robust and useful technique that predicts the best interaction between ligands and target and ranks the ligands according to their affinity for the target site [20].

Another crucial computational method in this study is molecular dynamics (MD). MD is a computational simulations technique employed in the study of biological molecules to analyze the physical behavior of the constituent atoms and molecules [21]. MD simulations serve as an invaluable tool that gives insight into the structural changes of the protein at different time scales [22]. The receptor, ligand, and overall complex motions, which are vital information provided by MD simulations can be exploited for drug design processes [22].

The main goal of the present work is to apply the drug repurposing approach and to identify potential inhibitors against FadD32. This approach is focused on virtual screening (VS), molecular dynamics (MD), simulations and post-MD calculations. In addition, the study provides a molecular understanding of the surrogate structure (FadD32) and how it interacts with a ligand from a computational standpoint. Hence, the current study applied VS approach and identify four potential (Figure 2) FDA-approved hits (accolate, sorafenib, mefloquine, and loperamide).

We believe the approach ‘drug repurposing’ could be carried out in the procedure of drug discovery of potential drugs against a wide range of biological targets. To the best of our knowledge, this is the first time where computational tools have been applied to reveal the impact of FadD32 upon the binding of FDA-approved drugs.

## 2. Results and Discussion

### 2.1. Molecular Docking and Binding Free Energy

All four hits (accolate, sorafenib, loperamide, and mefloquine) were subjected to docking followed by MMPBSA calculation to determine the accuracy of the ligand–receptor binding affinities. As shown in Table 1, the docking scores for all four drugs ranged from −8.0 to −10.0 kcal/mol.

To gain insight into the binding free energy profiles, the MM-PBSA approach [23] was also carried out for all systems. The total binding energies of all the systems ranged from −45.13 to −21.52 kcal/mol, with accolate (−45.13 kcal/mol) and sorafenib (−32.73 kcal/mol) showing the highest total binding energy while mefloquine (−26.84 kcal/mol) and loperamide (−21.52 kcal/mol) showing the lowest total binding energy (Table 1). It was also observed that accolate also had the highest van der Waals and electrostatic contributions towards the total binding free energy (ΔE_vdw_ of −64.54 kcal/mol and ΔE_ele_ of −28.89 kcal/mol) and loperamide showed the lowest contributions (ΔE_vdw_ of −33.45 kcal/mol and ΔE_ele_ of −11.37 kcal/mol). Studies have shown that docking alone cannot deliver authentic results. Hence, results obtained from MMPBSA are more authentic than the energy contributions obtained from the docking calculations only.

### 2.2. Protein–Ligand Interaction Analysis

Per-residue energy decomposition calculation was carried out to gain insight on each amino acid residue contribution towards the binding [24], the results are as shown in Figure 3 and Figure 4. All the drugs bound in the N-terminal and C-terminal domain interface, an ATP binding site (Appendix A). Figure 3 displayed more hydrophobic interactions in the case of FadD32-Sorafenib (15) while FadD32-Mefloquine displayed the least hydrophobic interactions (6), and more hydrogen bonds were observed in the case of FadD32-Mefloquine (4) while FadD32-Sorafenib displayed the least hydrogen bonding (1). Hence, we can state that the total binding energy is influenced by the amount of energy contributed by each amino acid in the ligand binding not the number of amino acids bound to the ligand.

As shown in Figure 4, our results suggest that the highest residual energy contributions came from the following amino acids: Ile193 (−2.84 kcal/mol), Pro194 (−2.38 kcal/mol), and Phe625 (−2.09 kcal/mol) in the case of FadD32-Accolate, Tyr343 (−2.63 kcal/mol) and Arg192 (−1.73 kcal/mol) in case of FadD32-Sorafenib, Arg192 (−2.41 kcal/mol) in case of FadD32-Mefloquine, and Arg192 (−1.63 kcal/mol) in case of FadD32-Loperamide. Upon general observation, the van der Waals interactions showed significant contributions, towards the total binding energy, as compared to the electrostatic interactions (Figure 4). The best amino acid energy contributors form hydrogen bonds with the ligands, except in the case of FadD32-Sorafenib and the Phe625 in the case of FadD32-Accolate (Figure 3). These are further supported by the hydrogen bonding data presented in Appendix A.

The results herein showed that the hit accolate drug formed more interactions with FadD32 active site residues as compared to the mefloquine, sorafenib, and loperamide. In addition, in the case of accolate, all active site residues displayed an increase in total binding contribution. We believe that the presence of a sulphur atom, in the case of accolate which is absent in other hits, leads to higher binding affinities. Hence, current drugs such as thioacetazone, isoxyl, and ethionamide for the treatment of TB targeting the mycolic acid biosynthetic pathway possess sulphur atoms [25]. In this work, we present novel potential inhibitors of the *Mtb*-FadD32 bacteria as an alternative treatment for TB. Hence, our MMPBSA results strongly select the accolate hit as the most promising drug candidate for targeting *Mtb*-FadD32.

### 2.3. Structural Analysis

The root mean square deviation (RMSD) describes the structure’s conformational changes, by estimating the deviation in the Cα atoms of the residues on the backbone structure [26]. These conformational changes express the degree of protein stability. RMSD was calculated and the results are presented in Figure 5.

As shown in Figure 5, the RMSD of all systems were observed in the case of FadD32-Accolate, FadD32-Apo, FadD32-Mefloquine, FadD32-Sorafenib, and FadD32-Loperamide with an average of 1.05 Å, 1.20 Å,1.35 Å, 1.56 Å, and 1.63 Å, respectively. The average RMSD values of all systems ranged from 1.05 to 1.63 Å. Hence, all systems were found to be stable with the average RMSD lower than the ideal of 3.0 Å RMSD value [27].

### 2.4. Influence of the Drugs on FadD32 Amino Acids Mobility

The RMSF describes the dynamic behavior of individual amino acids within the structure, by estimating the Cα atoms fluctuations throughout the simulation [28]. These residue fluctuations express the degree of protein flexibility. To measure the degree of protein flexibility, RMSF was calculated, and the results are presented in Figure 6 with the average RMSF values ranging from 0.95 to 1.54 Å.

In Figure 6, the RMSF results show a similar trend as the RMSD where FadD32-Accolate demonstrates low RMSF with an average of 0.95 Å as compared to FadD32-Apo (1.04 Å), FadD32-Sorafenib (1.54 Å), FadD32-Mefloquine (1.27 Å), and FadD32-Loperamide (1.32 Å). Hence, our results suggest that the presence of accolate in the binding site reduced the mobility of amino acids as compared with sorafenib, mefloquine, and loperamide. In addition, in the case of the apo system, the most notable changes can be seen in the following regions: Arg269-Gly275 and Ile538-Asp543 showing higher fluctuation. However, the presence of accolate reduced the fluctuation in these regions.

Our results conclude that the FadD32 protein is highly flexible during the process of biosynthesis of mycolic acids, essential for the growth and multiplication of the *Mtb.* However, the presence of accolate in the binding site leads to conformational rigidity. These findings are in correlation with the RMSD results that suggest lower system stability (RMSD: 1.05 Å) in the case of FadD32-Accolate.

### 2.5. Radius of Gyration (Rg)

In recent years, the radius of gyration has been applied to give insight into the level of compactness of the protein structure throughout the simulation [29]. To measure the level of compactness of the protein structure, Rg was calculated, and the results are presented in Figure 7 with the average Rg values ranging from 24.63–25.73 Å.

Figure 7 showed that, throughout the simulation, the FadD32-Accolate system displayed a lower Rg with an average of 24.63 Å, whereas FadD32-Apo (24.94 Å), FadD32-Sorafenib (25.73 Å), FadD32-Mefloquine (24.87 Å), and FadD32-Loperamide (24.91 Å) displayed a higher Rg. These results suggest that the presence of accolate into the FadD32 protein exerts conformational stability and compactness within the protein as compared to the rest of the ligands. The calculated Rg results correlate with the estimated RMSD and RMSF, which justified increased biomolecular flexibility of FadD32 protein in the absence of accolate. Our results show that the FadD32 protein appeared to be highly affected by the presence of accolate.

### 2.6. Dynamic Cross-Correlation Matrices (DCCM)

DCCM is a 3D matrix representation that gives insight on time-correlated residue motions of protein of interest [30]. DCCM analysis was conducted, and results are presented in Figure 8. The red-orange regions (0.5–1.0) represent strongly correlated/positive motions, whereas the yellow regions (0.25–0.50) represent slightly correlated motion, the green regions (0.25 to −0.25) represent regions with no correlation motions (no movement) and the blue-light blue regions (−0.50 to −1.0) represent strongly anti-correlated/negative motions.

Figure 8 displayed a positive correlation trend and a negative correlation trend at 1–250 and 500–630 residues, respectively, for all systems. Upon observation, the binding of all drugs introduced different dynamic changes within the FadD32 protein. The region of 1–250 relative to 1–250 residues in the case of FadD32-Accolate and FadD32-Sorafenib displayed the most strongly correlated motions, these are increased correlated motions when compared to FadD32-Apo. FadD32-Mefloquine demonstrated slightly correlated motions in this region. The region 480–630 relative to 1–250 residues, in the case of FadD32-Accolate, has strongly anticorrelated motions as compared to the other systems. This can be due to numerous ligand interactions that occur in this region. Hence, there is reduced flexibility, and this aligns with the RMSF results (Figure 7). In the same region, FadD32-Sorafenib and FadD32-Loperamide matrices demonstrated anticorrelated motions and FadD32-Mefloquine displayed partially anticorrelated motions with patches of no correlation motions. The internal region, opposite the latter region, displayed anticorrelated motions with variant intensities for each system. These findings have shed light on the investigated drugs and the FadD32 protein.

### 2.7. Principal Component Anaylisis (PCA)

PCA is a technique used to understand complex motions and flexibility within a protein, in the presence and absence of a ligand or inhibitor [31]. The conformational changes are measured by the directional eigenvalues, PC1 vs. PC2. The MD trajectories of all the systems were subjected to PCA calculations, taking to account the Cα atoms of residues and the results are depicted in Figure 9.

From the scatter plot in Figure 9, all the systems demonstrate different protein motions, the phase space occupied differs in the case of each system. FadD32-Accolate shows a pattern sort of similar to the FadD32-Apo, while occupying a smaller phase space and more compact. This proves that FadD32-Accolate exhibits lower molecular fluctuations as compared to the other systems, hence these results are consistent with the RMSD, RMSF, and Rg findings which stated that the binding of accolate reduced mobility on FadD32 residues and confers stability with the protein.

### 2.8. In Silico ADME Predictions

ADME predictions predict the nature, behavior, and fate of pharmaceutical drugs in an organism’s body. These molecular physicochemical parameters are essential in drug design and drug approval. The drugs (accolate, sorafenib, loperamide, and mefloquine) were evaluated on SwissADME [32] web server based on Lipinski’s rule of five and the results are presented in Table 2 and Table 3. Lipinski’s rule states that an orally active drug has no more than one violation of the following rules; molecular weight ≤500 g/mol, number of hydrogen atom donors ≤5, number of hydrogen atom acceptors ≤10, and the lipophilicity, Log *p* ≤ 4.15 [33]. All the drugs satisfied Lipinski’s rules (Table 2); hence, they have accepted drug absorption and permeation. This is further supported by the high bioavailability scores.

The results, described in Table 3, revealed that accolate and sorafenib had low human gastrointestinal absorption (HIA) and were not the substrate of P-glycoprotein (P-gp) while Mefloquine and Loperamide were found to be P-gp substrates and had high HIA. All the drugs have no blood–brain barrier (BBB) permeability except loperamide. Drug metabolism via CYP enzymes demonstrated variant results. All the drugs demonstrated positive ADME properties which suggest great promise for future treatment of TB. In addition, the usage and adverse effects of all selected FDA-approved drugs are given in Table 4.

## 3. Methodology

### 3.1. Computational Procedure

The X-ray crystal structure of *Mtb*FadD32 (PDB:ID 5HM3) presented in the literature bears major discrepancies [11]. Hence, *Mycobacterium smegmatis* FadD32 structure was applied as a surrogate to evaluate drug leads against *Mtb*FadD32. *M. smegmatis* (*Msm*) is the frequently used model for *Mtb* as it is a good device for studying the properties of mycobacteria [42]. In addition, *Msm*FadD32 shares 74% sequence similarity with *Mtb*FadD32 (Figure 10) [43], and most importantly *Msm* and *Mtb* show similar TB drug susceptibility [44].

### 3.2. System Preparation

In preparation for the molecular docking, Protein Data Bank (PDB) was used to obtain the X-ray structure of *Msm*FadD32 protein (PDB ID: 5D6J) [24]. The small molecules were retrieved from the ZINC database [45]. The UCSF Chimera [46] and Avogadro software [47] were used for the structural preparation of the *Msm*FadD32 receptor and the ligands.

### 3.3. Molecular Docking

The molecular docking tool, AutoDock Vina [48] was used to carry out the docking calculations. The docking process was executed using the default AutoDock Vina parameters. Blind docking was performed, the gridbox housed the entire protein receptor and these grid parameters were generated x = 88, y = 88 and z = 104 dimensions, x = −34.57, y = 4.91 and z = 11.35 centers with the exhaustiveness = 8. The docked conformations of the receptor–ligand complex were generated in a Lamarckian genetic algorithm approach in the order of their docking scores [48]. Docked complex conformations with the best docking score were visualized with UCSF Chimera [46], then considered for molecular dynamic simulations.

### 3.4. Molecular Dynamics (MD) Simulations

The MD simulations of the systems were carried out using the GPU version of the PMEMD module implemented in the Amber 14 software, with the Amber force field FF14SB [49] and general Amber force field (GAFF) [50]. Antechamber module was utilized to generate atom’s partial charges for ligands, hydrogen addition to protein, and system neutralization using the Leap module by adding the counter ions. The system was enclosed in a TIP3P water box, with a 10 Å distance between the system surface and box boundary. The system was subjected to initial minimization for 2500 steps and then heated gradually from 0 to 300 K with 1 ps, 5 kcal mol^−1^ Å^−2^ (collision frequency and harmonic restraints, respectively) settings using Langevin thermostat [51]. The system was equilibrated with no restrictions at 300 K,1 bar constant pressure, and the SHAKE algorithm [52] restricted the system’s bonds with hydrogen atoms. The system was subjected to a 150 ns MD in an isothermal-isobaric ensemble using Berendsen barostat, with 1 bar pressure and pressure-coupling constant of 2 ps. All other systems followed the same procedure.

### 3.5. Post-MD Simulation Analysis

After completing the 150 ns simulations, the MD trajectories were subjected to post-analysis calculations using the Amber14 modules PTRAJ and CPPTRAJ. These assist in the following analysis, MM-PBSA (molecular mechanics Poisson–Boltzmann surface area), Per-residue free energy decomposition analysis, hydrogen bonding analysis, RMSD (root mean square deviation), RMSF (root mean square fluctuation), Rg (radius of gyration) and DCCM (dynamic cross-correlation matrices).

The Equations (1) and (2) below describes how RMSD and RMSF are assessed:(1)$$\mathrm{RMSD}={\left(\frac{\sum_{N}{\left(R_{i}-R_{i}^{0}\right)}^{2}}{N}\right)}^{\frac{1}{2}}$$
where N is the number of Cα atoms in a complex, R_i_ is the position of vector of the Cα atom i, and $R_{i}^{0}$ is the position of vector for reference atom.
(2)$${\mathrm{sRMSF}}_{i}=\frac{{\mathrm{RMSF}}_{i}-\mathrm{RMSF}}{\sigma \left(\mathrm{RMSF}\right)}$$
where sRMSFi is the standardized RMSF, RMSFi is RMSF of the ith residue, RMSF average RMSF, and σ(RMSF) is the RMSF’s standard deviation.

### 3.6. Binding Free Energy Calculations and Per-Residue Free Energy Decomposition Analysis

Binding free energy calculation is a thermodynamics method that offers insight into the protein–ligand interaction [53]. In this study, the calculations were computed using the MM-PBSA approach, which is a popular method in drug design and estimates the interaction energy of protein–ligand (small molecules/inhibitor) complex [54]. All the system’s output trajectories were subjected to the calculation and the following Equations (3)–(6) describe the calculations:ΔG_binding_ = G_complex_ − [G_protein_ + G_ligand_](3)
ΔG_binding_ = E_MM_ + G_sol_ − TΔS(4)
ΔE_MM_ = E_ele_ + E_vdw_(5)
G_sol_ = G_polar_ + G_non-polar_(6)
where ΔG_binding_ denotes the protein–ligand complex’s free energy, E_MM_ is the sum of gas-phase molecular mechanics energy, G_sol_ denotes solvation free energy, and TΔS is total entropy. ΔE_MM_ is the sum of electrostatic and van der Waals contributions which are denoted by E_ele_ and E_vdw_, respectively. G_sol_ is the sum of polar and non-polar contributions, denoted by G_polar_ and G_non-polar_, respectively. The MM-PBSA approach was also used to determine the individual amino acid residue energy contributions towards the overall binding free energy. Protein–ligand interactions were visualized using LigPlot [55].

### 3.7. Dynamic Cross-Correlation Matrices (DCCM)

DCC between atoms is defined by the following expression:C_ij_ = 〈Δr_i_.Δr_j_〉/(〈Δr_i_^2^〉〈Δr_j_^2^〉)^1/2^(7)

In Equation (7), *i*th and *j*th denote the amino acids with their spatial backbone atom positions r_i_ and r_j_. Δr_i_ indicates the *i*th displacement from its mean position over time [56]. Each Cij element has a time scale associated with it, that correlates with a dataset of adjoining snapshot structures taken from the temporal succession of snapshot structures saved on the MD trajectory. The MD trajectories of different systems were subjected to the calculations and the matrices were generated, the Origin software [57] was used to analyze the atomic correlative motion results obtained.

### 3.8. Principal Component Analysis (PCA)

Principal components of the protein motion were calculated as described below:(8)$$C_{\mathrm{ij}}=\langle \left(x_{i}-\langle x_{j}\rangle \right)\rangle \;\left(i,j=1,\;2,\;3,\ldots ,3N\right),$$
where N is the number of Cα atoms, x_i_ and x_j_ are Cartesian coordinates of the *i*th and *j*th atoms respectively and ⟨x_i_⟩, ⟨x_j_⟩ denote the average time of all configurations of all the configurations obtained in MD simulation [58]. The MD trajectories of all the systems were subjected to the calculations and the Origin software [57] was used to draw the graphs for analysis.

### 3.9. In Silico ADME Predictions

The SwissADME web tool [32] was utilized for the assessment of absorption, distribution, metabolism, and excretion (ADME) parameters of the investigated drugs. This drug data is essential for drug approval, as it reveals the drug-likeness of the investigated drugs. The tool reveals the bioavailability of the drug candidates by estimating the following physicochemical properties: lipophilicity, size, polarity, solubility, saturation, and flexibility. The default predictors were used for this study.

## 4. Conclusions

Frequent development of drug resistance by *Mtb* against most approved TB drugs remains a major global threat, hence, motivates the urgent need for new effective drugs and new drug targets. In this report, we embarked on various computational approaches such as virtual screening (VS), molecular docking, and molecular dynamic (MD) simulation in order to identify potential FDA-approved drugs against fatty acid degradation protein D32 (FadD32), a novel drug target. The calculated ligand–protein binding energies ranged from −45.13 to −21.52 kcal/mol, with accolate (−45.13 kcal/mol) showing the highest total binding energy. It was observed that accolate also had the highest van der Waals and electrostatic contributions towards the total binding free energy (ΔE_vdw_ of −64.54 kcal/mol and ΔE_ele_ of −28.89 kcal/mol). Hence, the current study identified accolate as the best potential drug inhibitor of FadD32. Per-residue energy decomposition calculations suggest that the highest residual energy contributions came from Ile193 (−2.84 kcal/mol), Pro194 (−2.38 kcal/mol), and Phe625 (−2.09 kcal/mol) in the case of FadD32-Accolate. Therefore, the molecular structure of these residues will require careful consideration when designing inhibitors targeting FadD32. To provide insight into the structural, and mechanistic features of accolate as an FadD32 inhibitor, structural analysis was carried out by computing RMSD, RMSF, Rg, DCC and PCA. Accolate binding leads to FadD32 structural stability, hence reduced residue mobility and increased compactness of protein structure. Our results strongly suggest accolate as a potential inhibitor of *Mtb*FadD32, however, an experimental approach is required to validate the current hypothesis. We believe these findings will advance the design of potent *Mtb* inhibitors towards the treatment of TB. 

## Figures and Tables

**Figure 1 molecules-27-00668-f001:**
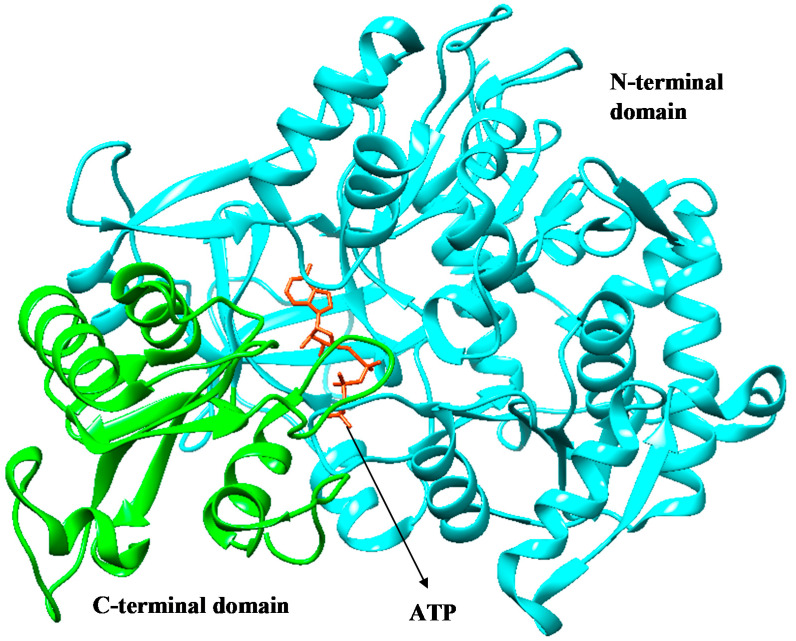
Three-dimensional (3D) structure of FadD32, showing the C-terminal domain (green) and the N-terminal domain (cyan), with the natural substrate, ATP (orange) [10].

**Figure 2 molecules-27-00668-f002:**
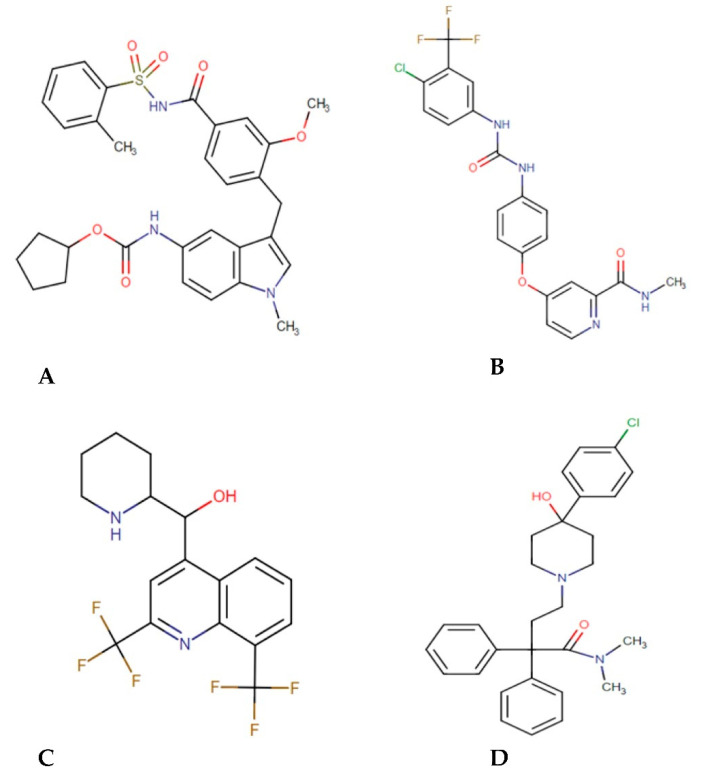
Chemical structures of accolate (**A**), sorafenib (**B**), mefloquine (**C**), and loperamide (**D**).

**Figure 3 molecules-27-00668-f003:**
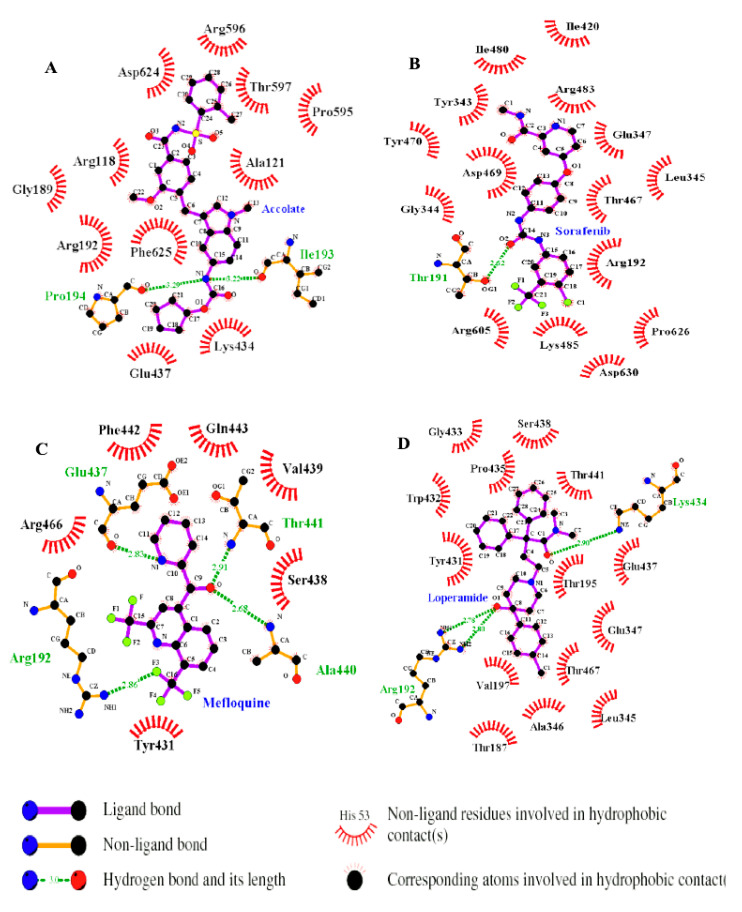
Interaction profile of FadD32-Accolate (**A**), FadD32-Sorafenib (**B**), FadD32-Mefloquine (**C**), and FadD32-Loperamide (**D**).

**Figure 4 molecules-27-00668-f004:**
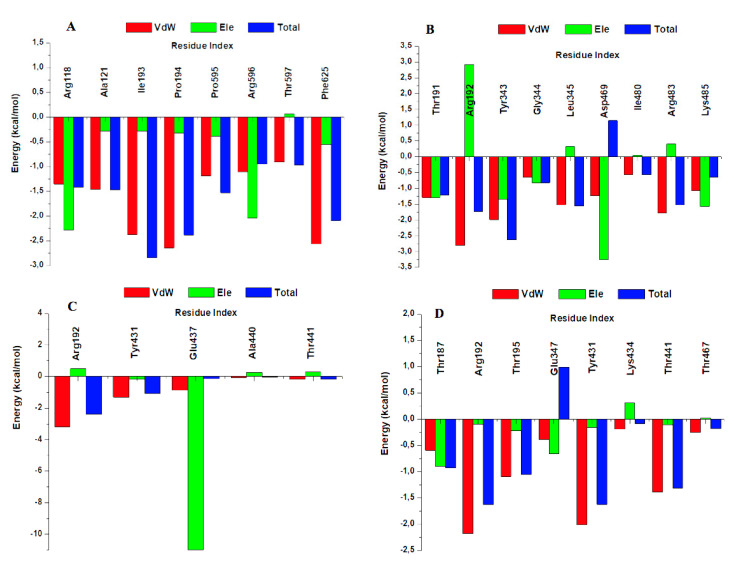
Per-residue decomposition analysis of FadD32-Accolate (**A**), FadD32-Sorafenib (**B**), FadD32-Mefloquine, (**C**) and FadD32-Loperamide (**D**).

**Figure 5 molecules-27-00668-f005:**
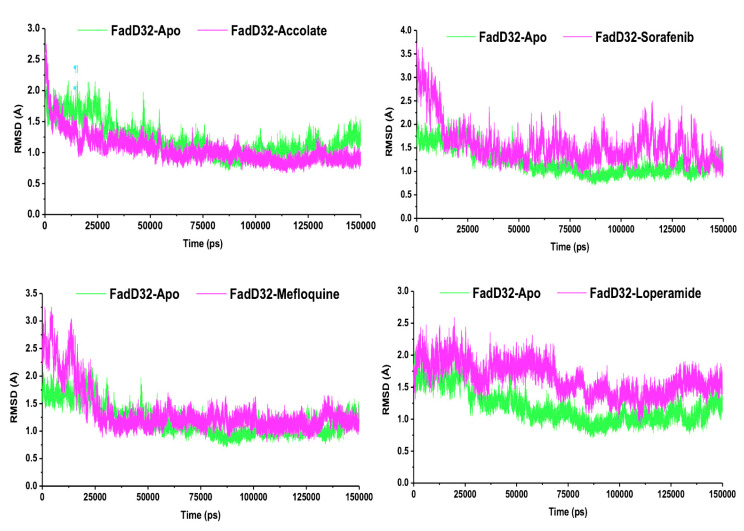
RMSD Plot of FadD32-Apo, FadD32-Accolate, FadD32-Sorafenib, FadD32-Mefloquine, and FadD32-Loperamide.

**Figure 6 molecules-27-00668-f006:**
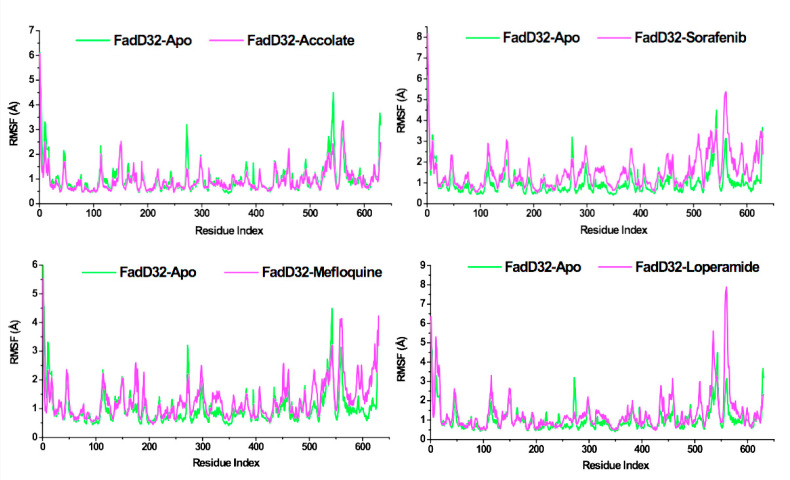
RMSF Plot of FadD32-Apo, FadD32-Accolate, FadD32-Sorafenib, FadD32-Mefloquine, and FadD32-Loperamide.

**Figure 7 molecules-27-00668-f007:**
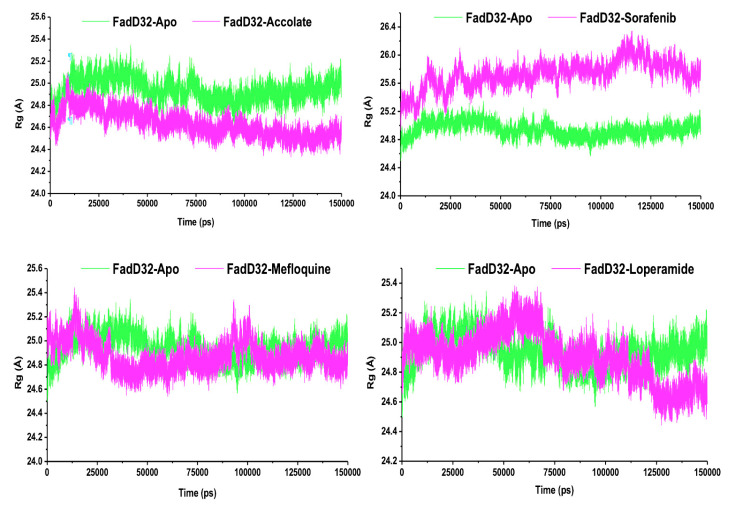
Radius of gyration plot of the FadD32-Apo, FadD32-Accolate, FadD32-Sorafenib, FadD32-Mefloquine, and FadD32-Loperamide.

**Figure 8 molecules-27-00668-f008:**
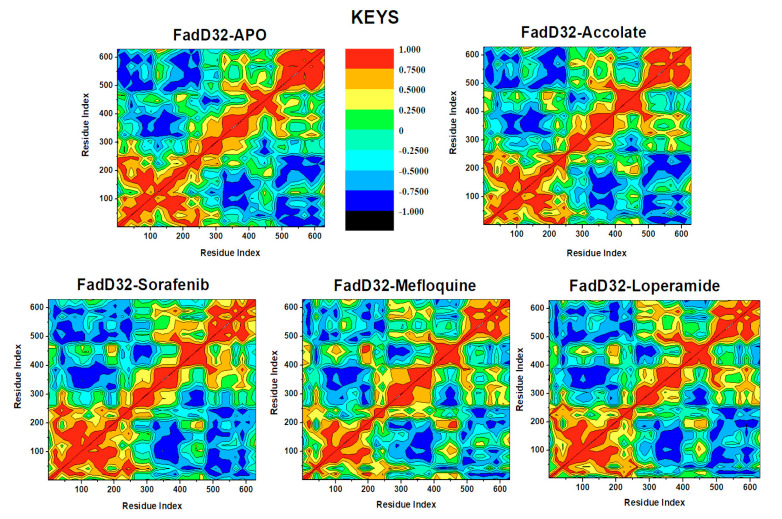
DCCM analysis graphs of the FadD32-Apo, FadD32-Accolate, FadD32-Sorafenib, FadD32-Mefloquine, and FadD32-Loperamide.

**Figure 9 molecules-27-00668-f009:**
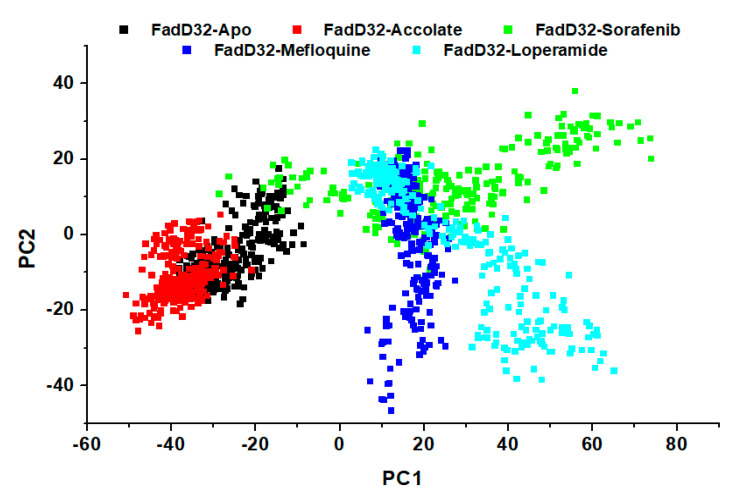
PCA analysis graph projecting the eigenvalues of FadD32-Apo, FadD32-Accolate, FadD32-Sorafenib, FadD32-Mefloquine, and FadD32-Loperamide along PC1 and PC2.

**Figure 10 molecules-27-00668-f010:**
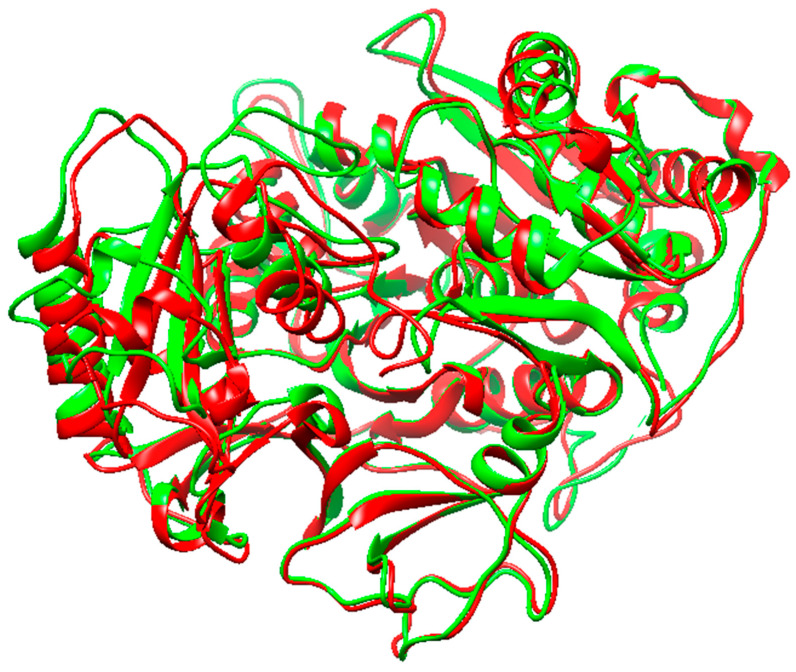
Aligned structures of *Mtb*FadD32 (green) and *Msm*FadD32 (red) proteins.

**Table 1 molecules-27-00668-t001:** Binding free energy (kcal/mol) and the components of binding free energy of the different molecules.

FDA-Approved Drugs	Docking Score (kcal/mol)	ΔG_bind_ (kcal/mol)	ΔE_vdw_ (kcal/mol)	ΔE_ele_ (kcal/mol)	ΔG_gas_ (kcal/mol)	ΔG_sol_ (kcal/mol)
Accolate	−9.3	−45.13 ± 6.64	−64.54 ± 4.08	−28.89 ± 9.70	−93.44 ± 12.41	48.31 ± 6.95
Sorafenib	−10.0	−32.73 ± 3.87	−51.64 ± 2.92	−27.74 ± 8.37	−73.37 ± 8.36	46.65 ± 7.83
Mefloquine	−8.0	−26.84 ± 2.63	−34.55 ± 2.74	−23.59 ± 6.18	−58.13 ± 6.15	31.29 ± 4.97
Loperamide	−8.5	−21.52 ± 7.40	−33.45 ± 7.95	−11.37 ± 7.48	−59.84 ± 10.21	28.39 ± 6.01

ΔG_bind_—Total binding energy; ΔE_vdw_—Van der Waals; ΔE_ele_—Electrostatic; ΔG_gas_—gas-phase energy ΔG_sol_—solvation energy.

**Table 2 molecules-27-00668-t002:** Physicochemical parameters for the drugs.

Parameters	Accolate	Sorafenib	Mefloquine	Loperamide
Bioavailability score	0.55	0.55	0.55	0.55
Molecular weight (g/mol)	575.68	464.82	378.31	477.04
Hydrogen bond donors	2	3	2	1
hydrogen bond acceptors	7	7	9	3
Lipophilicity (MLOGP)	3.92	2.91	3.43	4.17
Polarity: TPSA (Å^2^)	127.60	92.35	45.15	43.78
Lipinski violations	1; Mw > 500	0	0	1; MLOGP > 4.15

**Table 3 molecules-27-00668-t003:** Pharmacokinetics parameters for the drugs.

Parameters	Accolate	Sorafenib	Mefloquine	Loperamide
Gastrointestinal absorption	Low	Low	High	High
BBB permeant	No	No	No	Yes
P-glycoprotein substrate	No	No	Yes	Yes
CYP1A2 inhibitor	No	Yes	No	No
CYP2C19 inhibitor	No	Yes	No	No
CYP2C9 inhibitor	Yes	Yes	No	No
CYP2D2 inhibitor	No	Yes	Yes	Yes
CYP3A4 inhibitor	No	Yes	Yes	Yes
Log k_p_ (skin permeant) (cm/s)	−5.52	−6.25	−6.04	−5.65

**Table 4 molecules-27-00668-t004:** Usage and adverse effects of identified *Mtb*FadD32 potential inhibitors.

Drugs	Usage	Adverse Effects
Accolate	Treats and manage asthma in children (≥5 years old ) and adults [34]	Agitation, repetitive behaviours and progressive liver failure [35]
Sorafenib	Treats unresectable liver carcinoma and primary kidney cancer [36]	Diarrhea, fatigue and hypertension [37]
Mefloquine	Treats malaria infections [38]	Psychosis, convulsions and acute brain syndrome [39]
Loperamide	Treats and control nonspecific and chronic diarrhea [40]	constipation, drowsiness and abdominal discomfort [41]

## Data Availability

Data is contained within the article or Appendix A.

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
