# Peer review of "In Silico Drug Repurposing Approach: Investigation of Mycobacterium tuberculosis FadD32 Targeted by FDA-Approved Drugs"

_molecules, 2022, doi:10.3390/molecules27030668_

Round 1

Reviewer 1 Report

Interesting valuable work

Author Response

We would like to acknowledge and appreciate the editor and reviewers for taking their time in providing their insightful comments and suggestions. We have made a great attempt in addressing each and all points and concerns raised by reviewers. Our responses to the reviewer’s comments are provided in the table below. All changes and amendments are made in RED in the revised manuscript.

Reviewer accepted the manuscript (No revision required) 

Thanks and Regards

Authors

Reviewer 2 Report

The Authors provided a computational procedure to identify potential FDA-approved drugs against FadD32, a fatty acid degradation protein of Mycobacterium tuberculosis. Combining docking, Molecular dynamics (MD) simulations and MD based methods (such as MM-PBSA), they predicted potential drugs for tuberculosis therapy.

Four compounds were studied (Accolate, Sorafenib,Mefloquine and Loperamide) and four MD simulations - plus an Apo simulations - were performed to study basic protein behaviours, in presence/absence of the ligands.

I think that the paper is of interest, but it requires major revisions:

  1. Initial structure. The Authors explained that they used the X-ray structure of Mycobacterium smegmatis (Msm) instead of the structure of M. tuberculosis (with a 74% of seq similarity). They justified the choice saying that the structure of M.tuberculosis has missing amino acid residues. This explanation is not sufficient, because nowadays modelling missing residues is very easy. I checked both the structures and I think that the problem is in the C-terminal domain which is missing in the M.tuberculosis structure. As this domain is non-covalent bound, the only possible solution is to align the structures and create a “chimera”. Thus, considering this, the choice of taking the Msm structure makes more sense, but it needs to be specified, or it seems that the Authors are not able to rebuild missing residues.

Anyway, the PDB entry code of M.tuberculosis FsdD32 (PDB:ID 5HM3) has to be indicated, when they says (lines 103-105):

“The X-ray crystal structure of MtbFadD32 presented in literature (add PDB code) bears major discrepancies, numerous gaps, possesses non-standard amino acids and the structure information sheet is not in correspondence with the structure [11]”

Finally, a structural alignment between both the constructs  (5HM3 and 5D6J) could help in understanding similarity between the proteins, as well as the placement of missing residues. 

  1. The sampling must be increased. The Authors performed only one MD simulation of 150ns for each system. This is not sufficient. The Authors should perform at least 3 replicas for each system of a duration of at least 300ns, otherwise all their assumptions are lacking in scientific solidity.

  1. Analysis on the trajectories. The analysis provided can be improved. The DCC matrices are not well described, and in my point of view, the most important behaviours are not treated. 

First of all, DCC matrix is an analysis that allows to detect protein coupling motions. Thus, expressions such as “ demonstrates these as the most dynamic regions” (line 304-305, pag 12) are not appropriate. The only thing that we can deduce is that some regions correlate well, others anti-correlated, and so on. In fact, the presence of coupled motions is not necessarily a demonstration of the higher or less dynamic of the complex.

Anyway, the most important data is not the correlation within residues 1-250, but the anti-correlation between such residues and residues 500-600. In fact, these residues are far in the a.a sequences. Thus, my question is : are such residues also far away in the 3D structure? 

If yes, this means that opposite regions of the protein anti-correlates and this is more interesting than the correlation between the first 250 residues, that should be closer to each other.

Then, a picture underlining the main regions of the protein is fundamental to understand and interpret such data. 

  1. Additional analysis. An analysis that could help in understanding protein behaviours that is missing in this work is the principal component analysis (PCA). Several times the Authors say that the presence of Accolate in FadD32 “suppresses” fluctuations and flexibility of the protein.

Well, an analysis that evaluates conformational changes and it is useful to understand conformational differences due to presence/absence of a ligand is the PCA.

Thus, I suggest to perform a PCA on the combined trajectories (choosing alpha carbons), to project all the conformations in the same essential subspace. Then, the eigenvectors projections (maybe the only first 2 eigenvectors are enough to describe the motion, but it depends on your system) could also help in understanding which residues are more affected by the ligand interaction, in terms of increased/decreased mobility. 

If the Authors are not familiar with this analysis, following two of many papers about it:

Amadei, A., Linssen, A.B.M. and Berendsen, H.J.C. (1993), Essential dynamics of proteins. Proteins, 17: 412-425. https://doi.org/10.1002/prot.340170408

Martín-García F, Papaleo E, Gomez-Puertas P, Boomsma W, Lindorff-Larsen K (2015) Comparing Molecular Dynamics Force Fields in the Essential Subspace. PLoS ONE 10(3): e0121114. https://doi.org/10.1371/journal.pone.0121114”

Moreover, I suggest some minors - but really important - revisions:

  1. Figure 1 must be improved. It can be helpful for the reader to better understand the structure of the protein. Furthermore, the C-terminal part is a domain, and it is non-covalent  bound, but I realized this when I displayed the X-ray structure. This must be explained, or it seems that the protein is composed of only one covalent bound chain (this is important in terms of dynamic behaviours).

2.In the ”Methods” section:

  1. the number of simulations performed must be specify;
  2. the formulas for RMSD and RMSF should be reported.
  1. The title of paragraph 3.3 should be modified as “Structural Analysis”. Also, saying that ‘RMSD is a measure of the stability of the system’ may be ambiguous. It is better to say that the system reaches convergence (or equilibrium) after 30 ns (e.g. mefloquine complex).  Also, your proteins are likely out of equilibrium as the MDs are too short. However, the debate on the definition of when a protein reaches equilibrium is still open, but I strongly recommend increasing sampling and avoiding the use of ambiguous definitions.

  1. RMSF analysis. Please, specify which atoms were chosen for the calculations (alpha carbons, backbones etc..).

  1. pag 10 lines 262 and 266. Please, avoid the use of the word “suppress”, as the fluctuations are not suppressed but reduced.

  1. The “conclusions” must be improved by saying something more about the overall view of the work.

I think that if the Authors provide all these revisions, the quality of the paper will increase a lot.

Reviewer 3 Report

  1. The title of the manuscript is unnecessarily lengthy. Authors must consider a new and brief title for the manuscript.
  2. In the second line of the abstract, TB was mentioned as a major global pandemic. Using a pandemic statement for TB will be inappropriate. Authors can write it as “the disease remains a major global threat”.
  3. This sentence needs rephrasing “Fatty acid degradation protein D32 (FadD32), recently identified as a promising drug target against TB, is required for the biosynthesis of mycolic acids, hence, essential for the growth and multiplication of the mycobacterium.”
  4. This sentence needs rephrasing “Hence, our findings strongly suggest the Accolate hit as a potential drug, for TB therapy, against FadD32: a comprehensive binding analysis from molecular dynamic simulations.”
  5. “tubercular cell” is not the standard terminology. Please correct.
  6. This statement “Recently numerous potential inhibitors of Mtb drug targets have been identified through a drug repurposing strategy” should be accompanied by some examples.
  7. As the authors have used a surrogate protein from M. smegmatis, in replacement to MtbFadD32, authors should specify if the catalytic sites for both the proteins are same. Though they mentioned 74% similarity between MtbFadD32 and MsmFadD32, but catalytic site should be considered while selecting the surrogate.
  8. When I have checked the PDB site, I found that there are various ligand co-crystallized FadD32 PDBs are available. What is the reason for not selecting any of those, and preferring 5D6J.
  9. How many drug molecules were subjected to the virtual screening? What will be the rationale for drug selection?
  10. It seems like the authors have performed docking with only one hydrophobic cavity. What is the rationale for its selection? And how it will contribute to reducing the future susceptibility for the 4 drug hits to fall under resistance regimen like other already available drugs.
  11. As the four drugs are already FDA-approved drugs for specified therapeutic indication, so authors should add some content related to the existing usage and adverse effects reported for those drugs. For example, one of the rare side effects of Accolate is liver failure.  
  12. The authors should have used some known FadD32 inhibitors to compare the results. Refer to literature like https://www.ncbi.nlm.nih.gov/pmc/articles/PMC6233306/ ; https://www.pnas.org/content/110/28/11565 , etc.

Round 2

Reviewer 2 Report

The Authors provided all the suggested changes, and I think the paper is improved a lot. Thus, I recommend it for the pubblication in the current form.

Reviewer 3 Report

Authors have appropriately addressed most of my observations.